# Strategies for Positive Partial Transpose (PPT) States in Quantum Metrologies with Noise

**DOI:** 10.3390/e23060685

**Published:** 2021-05-28

**Authors:** Arunava Majumder, Harshank Shrotriya, Leong-Chuan Kwek

**Affiliations:** 1Indian Institute of Technology Kharagpur, Kharagpur 721302, India; 2Centre for Quantum Technologies, National University of Singapore, Singapore 117543, Singapore; harshank.s@u.nus.edu (H.S.); kwekleongchuan@nus.edu.sg (L.-C.K.); 3MajuLab, CNRS-UNS-NUS-NTU International Joint Research Unit, Singapore UMI 3654, Singapore; 4National Institute of Education, Nanyang Technological University, Singapore 637616, Singapore; 5Quantum Science and Engineering Center, Nanyang Technological University, Singapore 637616, Singapore

**Keywords:** PPT, quantum metrology, entanglement

## Abstract

Quantum metrology overcomes standard precision limits and has the potential to play a key role in quantum sensing. Quantum mechanics, through the Heisenberg uncertainty principle, imposes limits on the precision of measurements. Conventional bounds to the measurement precision such as the shot noise limit are not as fundamental as the Heisenberg limits, and can be beaten with quantum strategies that employ ‘quantum tricks’ such as squeezing and entanglement. Bipartite entangled quantum states with a positive partial transpose (PPT), i.e., PPT entangled states, are usually considered to be too weakly entangled for applications. Since no pure entanglement can be distilled from them, they are also called bound entangled states. We provide strategies, using which multipartite quantum states that have a positive partial transpose with respect to all bi-partitions of the particles can still outperform separable states in linear interferometers.

## 1. Introduction to Quantum Estimation Theory

Measurements underline all physical processes [1,2,3]. Yet, like all quantum processes, they is dictated by the laws of physics: Heisenberg-limited sensitivity. In optical interferometry, one such resource is the deployment of *N* maximally entangled probes in a NOON state [4,5,6].

Typically there are many quantities we cannot measure directly. A protocol for such measurements is typically indirect where we use an additional probe system to interact with the one under investigation. Due to interaction, the probe gains adequate information about the parameter we want to measure. We then inspect the probe arising from the measurement, and on the basis of the obtained data, we estimate the desired parameter. In order to obtain the highest possible accuracy for the estimation, we perform the experiment repeatedly a multiple of times or, equivalently, make a multipartite probe to interact with the system.

For a classical probe that contains *n* particles (we can also consider it a measurement performed *n* times) accuracy scales as $\frac{1}{\sqrt{n}}$, which is the so-called shot-noise limit (SNL). However, if the system is in a particular entangled state, then accuracy can be improved up to a precision of $\frac{1}{n}$. This limit, called the Heisenberg limit (HL), provides the best accuracy allowed by quantum mechanics. Both of these bounds can be derived from the quantum Cramer-Rao bound with quantum Fisher information (QFI).

Quantum entanglement [7,8] has been an extremely useful resource for numerous applications. Reference [3] reviews quantum metrology using many well known examples of entangled states such as the GHZ state, Dicke state and singlet state and alo describe the highest achievable precision using these states. The Peres-Horodecki theorem [9,10] provides a criterion for determining the entanglement of quantum states. If the partial transpose of a mixed state has negative eigenvalues, then the state is entangled. However, in general, we cannot say anything about the entanglement of a state if it does not have negative eigenvalues. It turns out that it is possible in $d\times d^{\prime }$ ($d,d^{\prime }\geq 3$) dimension for the quantum state to be entangled even if the eigenvalues of the partial transpose are all positive. Moreover, these states cannot be distilled to a singlet form and they are also known as bound entangled or positive partial transpose (PPT) states. However, bound or PPT states cannot be distilled to singlets, so these states are not suitable as a resource for quantum communications. Yet, these states can be pumped into Werner state (an admixture of singlet and maximally mixed state) to increase its singlet fractions [11]. Moreover, PPT states can enhance the ability for teleportation of other states [12] and they are found to be useful for quantum metrology [13,14]. The paper [14] shows that there are PPT states that are not directly useful for metrology. However, their metrological usefulness can be activated by using two or more copies of these states for the activation. In [15] the authors provided an inequality based on QFI which must be satisfied by separable states and violation of that inequality can immediately detect bound entangled or freely entangled states, e.g., states found in [13] will violate that inequality. In [16] the authors found another class of PPT states with a high Schmidt number for general d×d dimension states with d≥4, the metrological usefulness of such states are yet to be found.

In Section 2, we briefly describe Quantum Fisher Information (QFI) along with some useful properties. Section 3 compares classical and quantum strategies for metrology where the quantum strategies take advantage of entanglement by employing the use of entangled probes for improving QFI. In Section 4 we discuss at length the effect of using sequential strategies in improving QFI for the PPT states found in reference [13]. Next, in Section 5 we apply the sequential ancilla assisted strategy to a family of even dimensional PPT states from [17] and draw some inferences based on the improvement obtained in QFI. In Section 6 we apply the sequential approach to both entanglement and ancilla assisted strategies for the case of 3×3 and 4×4 PPT states and find that this approach gives an improved QFI for the cases considered. Finally, we discuss some implications of our results in Section 7.

## 2. Quantum Fisher Information

Fisher information was originally introduced for statistical estimation purposes. The classic Cramér-Rao inequality places an upper bound on the precision of the estimation in terms of Fisher information. Alternatively, this may be interpreted as that Fisher information has a lower bound in terms of the precision (variance) of any measurement. Fisher information has been generalized to quantum situation, and there is also a quantum analogue of the Cramér-Rao inequality [18,19,20].
(1)$${\left(\Delta \theta \right)}^{2}\geq \frac{1}{F_{Q}\left(\rho ,\hat{A}\right)}$$
where $F_{Q}\left(\rho ,\hat{A}\right)$ is the Quantum Fisher information (QFI). The quantum Fisher information, a central quantity in quantum metrology, is defined by the formula
(2)$$F_{Q}\left[\rho ,\hat{A}\right]=2\sum_{k,l}\frac{\left|\langle k\right|\partial_{\theta }\rho_{\theta }{\left|l\rangle \right|}^{2}}{\left(\lambda_{k}+\lambda_{l}\right)}$$

Equation (Equation 2) can also be written in the form below using Baker–Campbell–Hausdorff formula
(3)$$\rho \left(\theta \right)=U\left(\theta \right)\rho \left(0\right)U{\left(\theta \right)}^{†}=\rho_{in}-i\theta \left[A,\rho_{in}\right]$$

So, Equation (Equation 2) becomes
(4)$$F_{Q}\left[\rho ,\hat{A}\right]=2\sum_{k,l}\frac{{\left(\lambda_{k}-\lambda_{l}\right)}^{2}}{\left(\lambda_{k}+\lambda_{l}\right)}{\left|\langle k|A|l\rangle \right|}^{2}$$
where, A = $\sum_{n=1}^{N}a^{\left(n\right)}$ and $a^{\left(n\right)}$ is the single particle operator acting on the *n*th particle of the probe system also known as the unbiased estimator. $\lambda_{k}$ and |k〉 are the respective eigenvalues and eigen vectors of the density matrix ρ, that is used as a probe state for estimating θ.

For qubits, the $a^{\left(n\right)}$’s operators are just the spin operators in the *z*-direction, $\sigma_{z}/2$ so that A = $\sum_{n=1}^{N}\sigma_{z}^{\left(n\right)}/2$ but for d>2, i.e., qudits A = $\sum_{n=1}^{N}D^{\left(n\right)}$, where $\sigma_{z}^{\left(n\right)}$’s are traditional Pauli operators and D = Diagonal [1,1,...,−1,−1], for even ’d’ there will be d/2 (+1)’s and d/2 (−1)’s and for odd ‘d’ there are $\frac{d+1}{2}$ (+1)’s and $\frac{d-1}{2}$ (−1)’s, e.g., for a bipartite(3×3) qutrit state
(5)$$A=D⊗I+I⊗D=A_{1}+A_{2},$$
where I=Identitymatrixofdimensiond×d, and $A_{1}$ and $A_{2}$ are called local Hamiltonians.

For separable multi-qubit states, the quantum Fisher information, characterizing the maximal precision achievable by a quantum state, is bounded as
(6)$$F_{Q}\left[\rho ,\hat{A}\right]\leq N$$
where *N* is the number of subsystems.

For separable multi-qudit (d>2) states, the maximum quantum Fisher information is bounded as
(7)$$F_{Q}\left[\rho ,\hat{A}\right]\leq \sum_{n=1}^{N}{\left[\lambda_{max}\left(a^{\left(n\right)}\right)-\lambda_{min}\left(a^{\left(n\right)}\right)\right]}^{2}$$
where $\lambda_{max}$ and $\lambda_{min}$ are the maximum and minimum eigenvalues of $a^{\left(n\right)}$.

Quantum Fisher information possesses some interesting mathematical properties [18]. Indeed, it has been shown that, amongst all generalized analogues of quantum Fisher information, the quantum Fisher information is the largest for rank-2 density matrices and for operators that have zero diagonal elements in the eigenbasis of the density matrix [21,22]. Here, we list some of the properties below [23,24]:For pure state ρ the QFI is given by $F_{Q}\left[\rho ,J_{l}\right]=4{\left(\Delta J_{l}\right)}_{\rho }^{2}$.$F_{Q}\left[\rho ,J_{l}\right]$ is convex in the state ρ, i.e., $F_{Q}\left[p_{1}\rho_{1}+p_{2}\rho_{2},J_{l}\right]\leq p_{1}F_{Q}\left[\rho_{1},J_{l}\right]+p_{2}F_{Q}\left[\rho_{2},J_{l}\right]$.$F_{Q}\left[\rho ,\hat{A}⊗I,I⊗\hat{B}\right]=F_{Q}\left[\rho ,\hat{A}\right]+F_{Q}\left[\rho ,\hat{B}\right]$.For *N*-qubit separable states, the values of $F_{Q}\left[\rho ,J_{l}\right]$ for l=x,y,z are bounded as
(8)$$\sum_{l=x,y,z}F_{Q}\left[\rho ,J_{l}\right]\leq 2N$$
in Equation (Equation 3) the range is given for one of the *l*’s among x,y,z.For Greenberger-Horne-Zeilinger states (GHZ states) (maximally entangled states), the quantum Fisher information is bounded by
(9)$$\sum_{l=x,y,z}F_{Q}\left[\rho ,J_{l}\right]\leq N\left(N+2\right)$$For *N*-qubit *k*-producible states states, the quantum Fisher information is bounded by
(10)$$F_{Q}\left[\rho ,J_{l}\right]\leq nk^{2}+{\left(N-k\right)}^{2}$$

## 3. Classical vs. Quantum Strategy

As shown in Figure 1, conventional measurement schemes (upper panel) prepare and detect *N* independent physical subsystems separately. The final result comes from a statistical average of the *N* outcomes. In quantum enhanced measurement schemes (lower panel), the physical system is prepared in a highly correlated state (i.e., an entangled or a squeezed state, here we have used PPT correlated) of *N* physical subsystems, and we measure collectively with a single non-local measurement that includes all the subsystems.

## 4. Sequential Strategies

In this section, we study the usefulness of the PPT two-qudit states introduced in reference [13] for quantum metrology. These states were shown to be useful for quantum metrology through a semidefinite program, which is essentially a convex optimization algorithm where the constraints are ρ≥0 and $\rho^{Tk}\geq 0$ for all *k* and Tr(ρ)=0. We first discuss the case in which both the qudits in the input PPT states undergoes a phase transformation so that the qudits in state are imprinted with a phase, and then we subject the state to two specific cases: (A) and (B) in Figure 2. Cases (A) and (B) have also been studied in reference [13] for maximally entangled states. In case (B), the authors in reference [13] studied a noisy state of the form
(11)$$\epsilon_{AB}\left[\rho \left(\theta \right)\right]=\rho \left(\theta ,p\right)=\left(1-p\right)\rho +p\frac{I}{d},$$
where $\epsilon_{AB}$ is the total depolarizing noise acting on both the subsystems of Figure 2B. If the same noisy channel acts repeatedly *m* times to the intermediate output state $\rho_{out}$ before the measurement then the final output state becomes
(12)$$\epsilon_{AB}^{⊗m}\left[\rho_{out}\right]=\rho_{out}\left(O\left(p^{m}\right),\theta \right).$$
where ρ is the d×d PPT state and *p* is the strength of noise. It has been shown that as the dimension *d* increases, the Fisher information increases (even though the Fisher information for a separable state remains at 8, and also that the noise resistance increases, i.e., the critical *p*, when the Fisher information falls below 8, increases as shown in Table 1 [13].

In Figure 3, we present a change in the strategy from the left one (entangled assisted strategy) to the right (sequential ancilla assisted strategy)) where we show that it is possible to increase the advantage by using sequential strategy. We also present the Table 1 where we apply the sequential strategy to the 3×3 to 12×12 PPT states in reference [13] and compute Fisher information for each of the states corresponding to both of the cases of Figure 3.

Here, the phase is imprinted using the unitary evolution on each of the qudits $U\left(\theta \right)=e^{-iD\theta }$ where D=diagonal[1,1,...,−1,−1] where for even dimensions, there are an equal number of −1 and 1 but for odd dimensions, the number of 1 is exactly one more than the number of −1. It is very important to notice that all U(θ)’s are local unitary operations acting on each of the subsystems independently, i.e., in the left figure on both the subsystems the local unitary operator acts as U(θ)⊗I (for the upper subsystem) and I⊗U(θ) (for the second one) respectively.

In the right figure, the total operator in Equation (Equation 5) is changed to
(13)$$A=D⊗I+D⊗I=A_{1}+A_{2}=A_{i}+A_{i}$$
where $A_{1}$ and $A_{2}$ are now same ($A_{i}$) in this case.

For experimentally implementing the sequential strategy using the operator (Equation 13), the unknown phase parameter (θ) can be pictured as being applied twice to just one of the two subsystems of the input probe through local unitary U(θ)⊗I, unlike the case of Equation (Equation 5), where the parameter (locally) is inserted at both the subsystems of the input probe. The operator ‘A’ can be broken as $A_{1}$ and $A_{2}$ being applied separately to each probe as in Equation (Equation 5) or being applied to the same probe as in Equation (Equation 13). This is mathematically equivalent to two local unitary operations U(θ)⊗I and I⊗U(θ) acting on the input probe in the left figure of Figure 3 and two consecutive local unitary operations U(θ)⊗I and U(θ)⊗I acting on the input probe in the right figure of Figure 3.

In the figures below, we show that we can apply our sequential strategy to both the left and right cases of Figure 2.

Figure 4A,B shows a sequential process of phase imprinting without (A) and with noise (B) performed repeatedly for *m* iterations.

Figure 5 shows the ancilla assisted strategy in which the phase is imprinted once (A) and *m* times (B) on the state before measurement. The Fisher information in Figure 5A in which the phase is encoded just on one of the qudit is a lot less than the Fisher information for a separable state. However, we can perform the sequential strategy to boost the quantum Fisher information as in Figure 3.

To see why, we find the QFI in which the phase is encoded *m* times. If we denote the Fisher information for just one iteration, $F_{Q}\left[\rho ,\hat{A}\right]$, as $F_{1}$, then the Fisher information after the second iteration is $F_{2}=4F_{1}$. In this way, if we iterate the same process *m* times then the Fisher information after the *m*th iteration becomes
(14)$$F_{m}=m^{2}F_{1}$$

Suppose we can now introduce noise in the system. If the noise acts just one subsystem in a bipartite qudit (d>3) system, it turns out that the noise model is not so trivial. This is not the case for d=2. The one-qubit Pauli channel for a bipartite qubit system can be written as
(15)$$\begin{array}{ccc} \epsilon_{A}\left[\rho \right] & = & \left(1-3p/4\right)\rho +p/4(\left(\sigma_{x}⊗I_{2}\right)\rho \left(\sigma_{x}^{†}⊗I_{2}\right)\\ & & +\left(\sigma_{y}⊗I_{2}\right)\rho \left(\sigma_{y}^{†}⊗I_{2}\right)+\left(\sigma_{z}⊗I_{2}\right)\rho \left(\sigma_{z}^{†}⊗I_{2}\right)) \end{array}$$

When the dimension of the system increases beyond d≥3, the number of generators for SU(*d*) increases as $d^{2}-1$. In particular, for d=3, there are a total of 8 spin observables. For d>3, the channel can be extremely complex and it is no longer trivial to describe the channel in tersm of suitable Kraus’ operators. The detailed calculations about the noisy quantum channel is given in Appendix A.

In all the plots below, in Figure 6 we are basically using both Equations (Equation 12) and (Equation 14) combined with m=3 in each case and depolarizing noise in Equation (Equation 11) and the highlighted region inside the plots are the intersections of the three iterations. Finally we study how $F_{Q}$ varies with the noise parameter.
(16)$${\hat{\rho }}_{output}\left(O\left(p^{m},\theta \right),\hat{A}\right)=\epsilon_{AB}^{⊗m}\left[U_{m}\left(\theta \right)U_{m-1}....U_{1}\left(\rho_{input}\right)U_{1}^{†}...U_{m-1}^{†}U_{m}^{†}\right]$$

With *m* = 3. Now the noise strength is of the order $O(p^{m})$ in $\rho_{output}$ as the noise along with the phase is iterating m times. It can be realized analytically. In Equation (Equation 16) the $\rho_{output}$ takes that particular form because the noisy channel in Equation (Equation 11) and the unitary transformation $U_{m}=\exp\left(-iD\theta \right)$ commutes so the order between them doesn’t matter.

In the table below we compare our results (in 4th column) with the results(in 3rd column) from reference [13] with the same optimal states taken from reference [13].

From the Table 1 it is clear that not all PPT states from reference [13] give a higher QFI for the sequential assisted strategy as compared to the entangled assisted strategy (see column 3 and 4), however it can be seen that odd dimensional PPT states (along with the case of d=4) give higher QFI. Another thing to notice is that the classical Fisher information remains the same in each case. This observation is visually depicted in the bar plot Figure 7 below.

## 5. A Family of Even Dimensional PPT States Having a Higher Fisher Information for Sequential Ancilla Assisted Strategy Compared to Entanglement Assisted Strategy

In this section we highlight the advantage of using the sequential ancilla assisted strategy by showing that the family of bipartite even dimensional states recently introduced in reference [17] shows an improvement in the Fisher information as compared to the entanglement assisted strategy. The family of 2d×2d dimensional states is given as:(17)$$\begin{array}{cc} \rho_{F1} & =\frac{p_{1}}{2d^{2}}\sum_{i,j=0}^{d-1}(|00ij\rangle \langle 00ij|+|11ij\rangle \langle 11ij|)\\ \; & +\frac{p_{1}}{2d\sqrt{d}}\sum_{i,j=0}^{d-1}(u_{ij}|00ij\rangle \langle 11ji|+u_{ij}^{*}|11ji\rangle \langle 00ij|)\\ \; & +\frac{p_{2}}{2d}\sum_{i=0}^{d-1}(|01ii\rangle \langle 01ii|+|10ii\rangle \langle 10ii|)\\ \; & +\frac{p_{2}}{2d}\sum_{i,j=0}^{d-1}(u_{ij}|01ii\rangle \langle 10jj|+u_{ij}^{*}|10jj\rangle \langle 01ii|) \end{array}$$
with $p_{1}=\sqrt{d}/\left(1+\sqrt{d}\right)$, $p_{2}=1-p_{1}$ and $u_{ij}$ being the matrix elements of a unitary operator acting on a *d* dimensional space such that $|u_{ij}|=1/\sqrt{d}$. The authors, in [17], also noted that only the 4×4 PPT state used in [13] is related to the family of states $\rho_{F1}$ (for d=2) through a local unitary transformation. For this family $\rho_{F1}$, there are $d^{2}$ eigenvectors for the eigenvalue $\Lambda_{v}=p_{1}/d^{2}$ given by;
(18)$$|v_{ij}\rangle =\frac{1}{\sqrt{2}}(|00ij\rangle +\sqrt{d}u_{ij}^{*}|11ji\rangle ).$$

There are *d* eigenvectors for the eigenvalue $\Lambda_{w}=p_{2}/d$ given by;
(19)$$|w_{i}\rangle =\frac{1}{\sqrt{2}}(|01ii\rangle +\sum_{j=0}^{d-1}u_{ij}^{*}|10jj\rangle ).$$

All other eigenvectors are associated with the zero eigenvalue and they are orthogonal to the ones mentioned above. These set of eigenvectors comprises of vectors having a similar form as $|v_{ij}\rangle$ and $|w_{i}\rangle$ but with a sign difference, given by;
(20)$$|v_{ij}^{-}\rangle =\frac{1}{\sqrt{2}}(|00ij\rangle -\sqrt{d}u_{ij}^{*}\left|11ji\rangle \right)$$
and
(21)$$|w_{i}^{-}\rangle =\frac{1}{\sqrt{2}}(|01ii\rangle -\sum_{j=0}^{d-1}u_{ij}^{*}|10jj\rangle ).$$

According to their statement if we consider $A=D⊗I+I⊗D=\sigma_{z}⊗I_{2}⊗I_{d\times d}+I_{2}⊗\sigma_{z}⊗I_{d\times d}$ as the optimal operator for the family of states $\rho_{F_{2}}$ then we will have Fisher information $16d^{2}\lambda_{v}$, which is the maximum we can get.

However, it is easy to see that the sequential ancilla assisted operator A=D⊗I+D⊗I (for two successive iterations) gives the quantum Fisher information exactly $F_{Q}\left(A,\rho_{F1}\right)=16$ for the family $\rho_{F1}$ for all *d*, which clearly shows the advantage of using sequential ancilla assisted scheme. This observation matches the findings in Table 1 where we obtain a QFI of 16 for the 4×4 state; however, we do not find any improvement in Fisher information for other even dimensional PPT states of Table 1 which suggests that those even dimensional states are not related to the family $\rho_{F1}$ by any local unitary operation.

## 6. Applying Iteration to Both Entangled Assisted Strategy and Ancilla Assisted Strategy

As we stated earlier, the resources used in both the parts of Figure 3 are equivalent thus the next iteration of both these cases should also be equivalent as shown in Figure 8,

In Figure 8 note that the total number of phase imprinting should be the same in both schemes, which is this case is four. So, *m* phase iterations in entangled assisted strategy is equivalent to 2m phase iterations acting on one qudit in sequential ancilla assisted strategy in Figure 3 and Figure 8. If $F_{E}^{0}$ is the Fisher information for 1st iteration in ENT assisted strategy then after *m* iterations, we have
(22)$$F_{E}^{m}=m^{2}F_{E}^{0}$$

If $F_{A}^{0}$ is the Fisher information for 1st iteration (see Figure 5) in ancilla assisted strategy then after *m* iteration, we get
(23)$$F_{A}^{m}={\left(2m\right)}^{2}F_{A}^{0}=4m^{2}F_{A}^{0}$$

Equations (Equation 22) and (Equation 23) must be used when dealing with both the strategies of Figure 3 with multiple iterations and comparing the two cases after certain number of iterations. Next, we show two examples of the advantage of using iteration in both the entangled and ancilla assisted strategies of Figure 3 for 3×3 and 4×4 dimensional PPT states of reference [13].

### 6.1. 3 × 3 PPT State

The sequential entangled assisted scheme provides a substantial increase in QFI (>200 within 5th iteration) in both the cases. Thus, without increasing the number of subsystems (here it is two) and dimensions (d=3) we can still boost the Fisher information by sequentially iterating in both strategies as shown in Figure 3.

In Figure 9 for d=3 system, we compute the Fisher information for both the schemes with m≥1 iterations with $F_{E}^{0}=8.0085$ and $F_{A}^{0}=2.0691$ and these strategies correspond to Figure 4A for *m* = 1 and Figure 5A.

### 6.2. 4 × 4 PPT State

Here in Figure 10 we consider $F_{E}^{0}=9.3726$ and $F_{A}^{0}=4$ and these values correspond to Figure 4A for *m* = 1 and Figure 5A.

So, comparing the two schemes with the help of Figure 3, Figure 8 and so on, we finally conclude that the equivalent ancilla assisted scheme to the entanglement assisted scheme provides better precision when we add iteration to the picture and even though the advantage is not huge (for all the states) but still can provide better precision than the schemes provided in the past [13].

### 6.3. General Dimension

We note that this advantage varies with dimensions and it can increase more or remain same which completely depends on the dimension of the input PPT state. From the Table 1 it is observed that for PPT states with even dimension, except 4×4, i.e., 6×6, 8×8, 10×10 and 12×12, there is no advantage even if we increase the number of iterations.

## 7. Discussion and Conclusions

In this paper, we identified the advantages of iteration for improving the Quantum Fisher Information (QFI) in entanglement and ancilla assisted strategies for PPT entangled states. We implemented such sequential strategies for PPT states for various dimensions and found that QFI increases quadratically with number of iterations in the noiseless case as shown in Figure 9. In the Section 4, we show the effect of iterations in the noisy cases for 1, 2 and 3 iterations for the entangled assisted strategies.

Further, we reiterate the fact that sequential strategies are completely distinct from a repetition of the experiments through multiple times. Repeating the same experiment say *n* times gives the effective Quantum Fisher Information $F_{n}^{Q}=nF_{1}^{Q}$, where $F_{1}^{Q}$ is the Fisher information for the experiment performed at first time while the sequential strategies shown here give an effective QFI $F_{m}^{Q}=m^{2}F_{1}^{Q}$ after *m* iterations, where $F_{1}^{Q}$ is the Fisher information at first iteration (if we consider the circuit in Figure 2 as 1st iteration). This highlights the fact that iterating strategies give an advantage over repetition of the experiment. However, we can always combine the repetition and iteration simultaneously, in such case the total Fisher information after *m* sequence of unitaries and *n* repetition of the experiment $F_{Total}^{Q}=nm^{2}F_{1}^{Q}$.

So far we have discussed only PPT states that are useful for metrology, but still there are PPT states that have no use in precision measurement. One such example is Equation (Equation 4), known as the Horodecki state of [11], in reference [11] and parametrized as
(24)$$\rho_{\alpha }=\frac{2}{7}|\psi_{+}\rangle \langle \psi_{+}|+\frac{\alpha }{7}\sigma_{+}+\frac{5-\alpha }{7}\sigma_{-}$$
where $\psi_{+}$ is given by
(25)$$\psi_{+}=\frac{1}{2s+1}\sum_{i=0}^{2s}|i\rangle |i\rangle$$
s = spin of the system = 3 and the $\sigma_{+}$, $\sigma_{-}$ are given as
(26)$$\sigma_{+}=\frac{1}{3}(|1\rangle |0\rangle \langle 1|\langle 0|+\left|2\rangle |1\rangle \langle 2|\langle 1\right|+|0\rangle |2\rangle \langle 0|\langle 2|)$$
(27)$$\sigma_{-}=\frac{1}{3}(|0\rangle |1\rangle \langle 0|\langle 1|+\left|1\rangle |2\rangle \langle 1|\langle 2\right|+|2\rangle |0\rangle \langle 2|\langle 0|)$$
$$\rho_{\alpha }=\left\{\begin{array}{cc} \mathrm{Separable} & \;2\leq \alpha \leq 3,\\ \mathrm{Bound}\;\mathrm{entangled} & \;3<\alpha \leq 4\\ \mathrm{Free}\;\mathrm{entangled} & \;4<\alpha \leq 5 \end{array}\right.$$

The state in Equation (Equation 24) gives maximum Fisher information in the Bound entangled region (3<α≤4) for the operator in Equation (Equation 5), of F(ρ,A)≃ 4.05, which is less than that of separable state $F^{SEP}=8$. Even if we use sequential operator in Equation (Equation 13), the max F(ρ,A)≃ 5 and remains less than the seperable one.

## Figures and Tables

**Figure 1 entropy-23-00685-f001:**
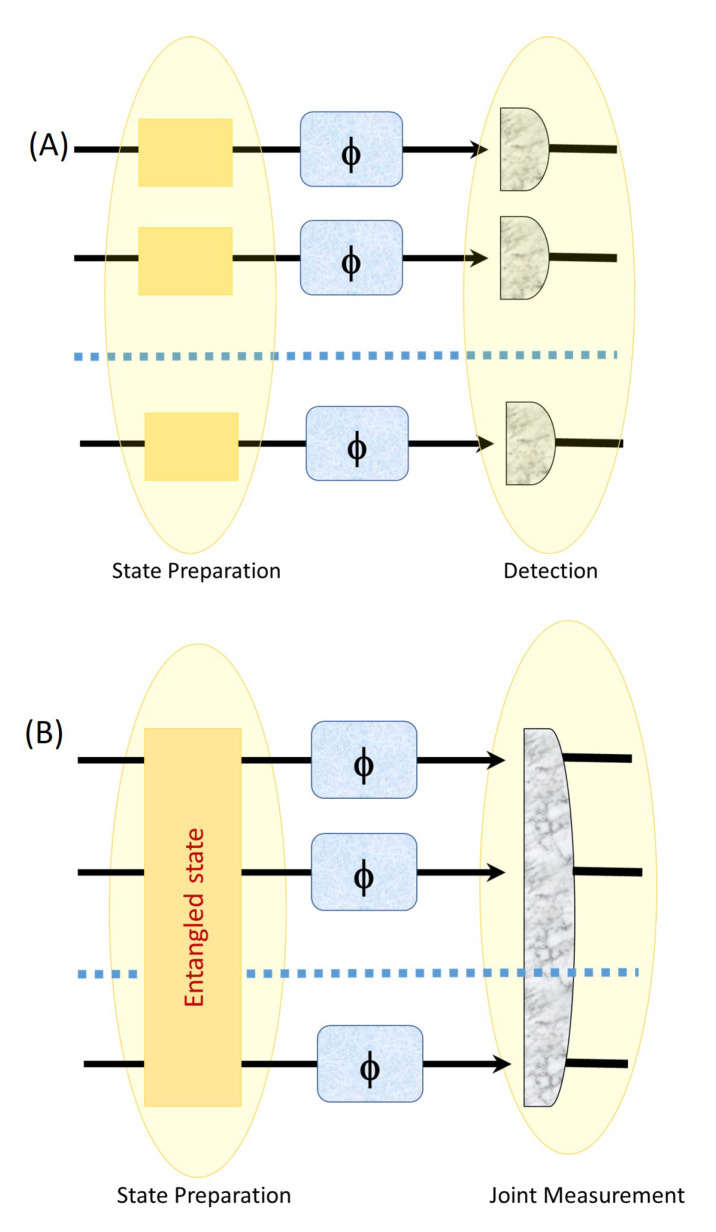
Classical (**A**) vs. Quantum strategy(Entangled assisted strategy) (**B**).

**Figure 2 entropy-23-00685-f002:**
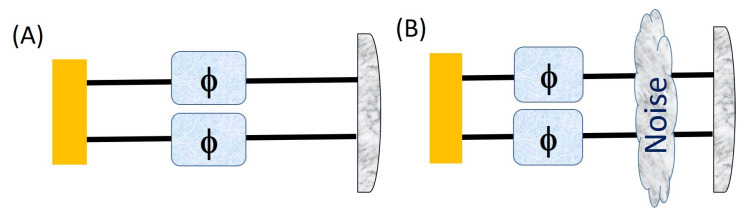
Two different strategies in which a state is imprinted with phase θ for each qudit and undergoes (**A**) no noise, and (**B**) “global” noise.

**Figure 3 entropy-23-00685-f003:**
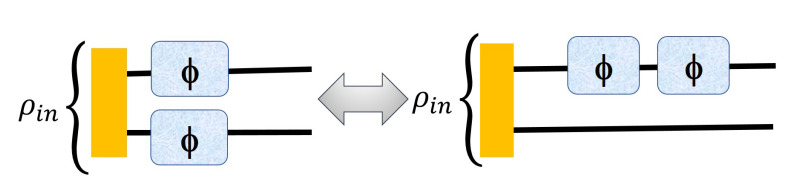
Change of strategy from entangled assisted strategy to sequential ancilla assisted strategy where the second qudit acts as an ancilla. However the right one has two iterations and thus two phases are imprinted on the state, just as in the left one. The sequential strategy (with just two iterations) in the ancilla assisted case (**right**) gives a better QFI compared to the entangled assisted strategy with one iteration (**left**), the amount of resources being the same in the two strategies.

**Figure 4 entropy-23-00685-f004:**
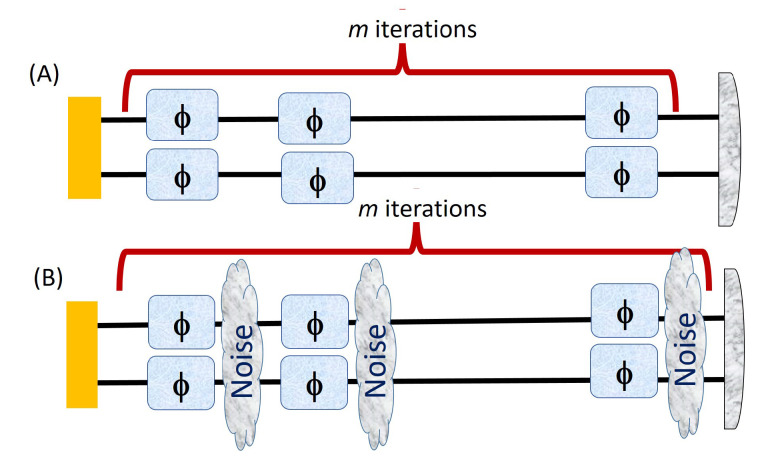
Two different strategies in which a state is imprinted with phase θ in both the qudits and undergoes the two cases as in Figure 2 but now the output state at one stage is fed into the input for *m* iterations. The subfigure (**A**) shows iterations without noise while subfigure (**B**) involves noise after each iteration.

**Figure 5 entropy-23-00685-f005:**
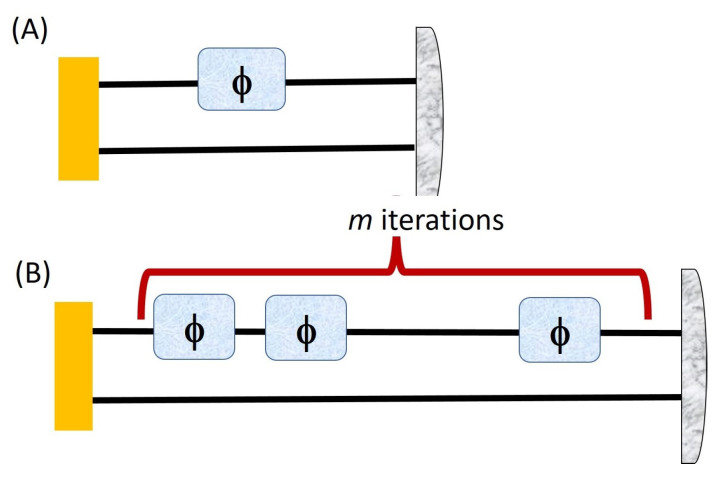
Two different strategies in which one qudit of a state is imprinted with phase θ and undergoes the iterated cases as in Figure 4. The (**A**) part is useless but the (**B**) with *m* = 2 is equivalent to Figure 4A (*m* = 1) thus we only have advantage from *m* = 2 ∼ Figure 4A (*m* = 1).

**Figure 6 entropy-23-00685-f006:**
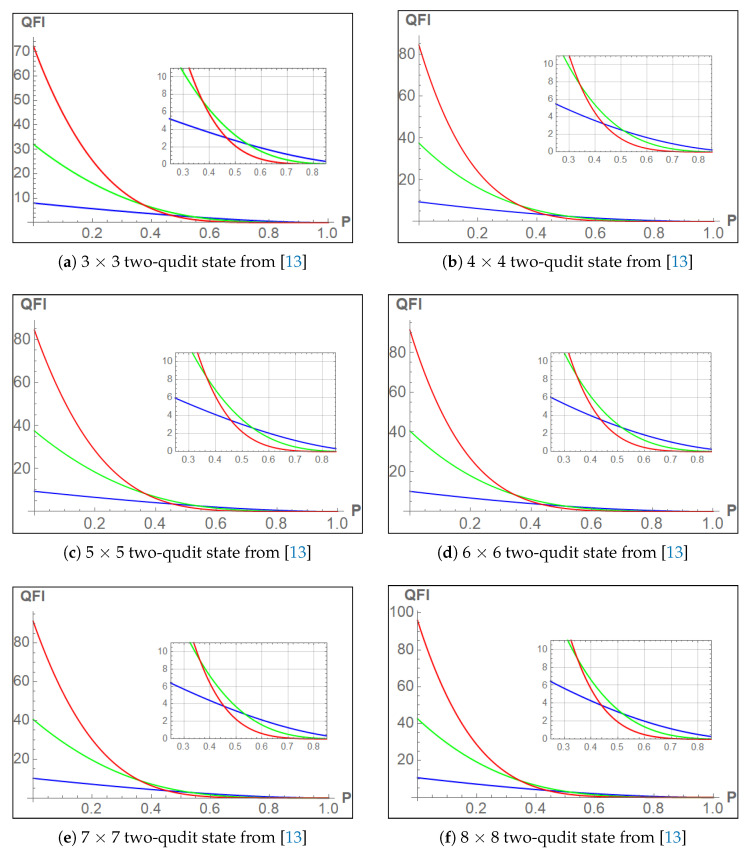
Quantum Fisher Information Vs Noise (P) plots (In each case blue, green and red are the 1st, 2nd and 3rd iterations respectively).

**Figure 7 entropy-23-00685-f007:**
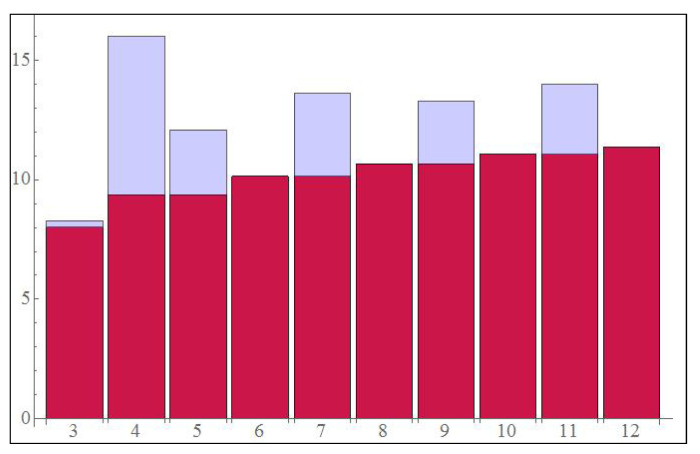
This figure shows the comparison between the two strategies of Figure 3. Dimension is varying along x-axis and QFI along y-axis. The Red bars correspond to the left strategy of Figure 3 and the blue ones correspond to the right one. Notice that the even dimensional states don’t provide any advantage of using sequential ancilla assisted strategy with two iterations instead of entangled assisted one with one iteration except 4×4 dimensional optimal state (for *m* = 2).

**Figure 8 entropy-23-00685-f008:**
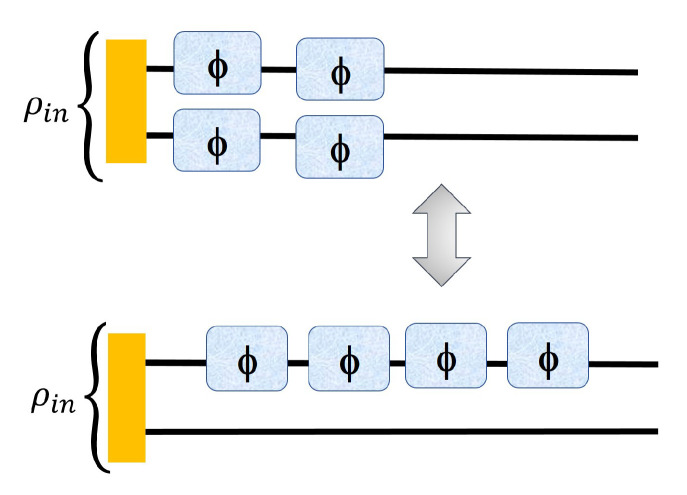
One more iteration applied to both the cases of Figure 3.

**Figure 9 entropy-23-00685-f009:**
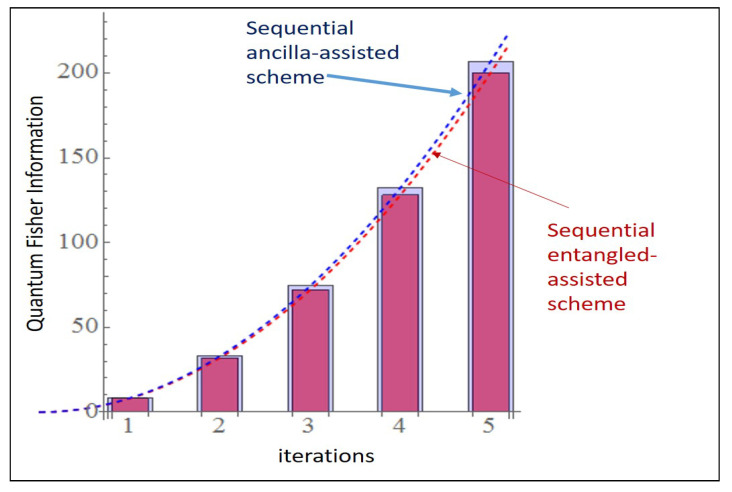
Comparison of two schemes in Figure 3 with multiple iterations for 3×3 the PPT state using Equations (Equation 22) and (Equation 23) combined as they are considered to be equivalent.

**Figure 10 entropy-23-00685-f010:**
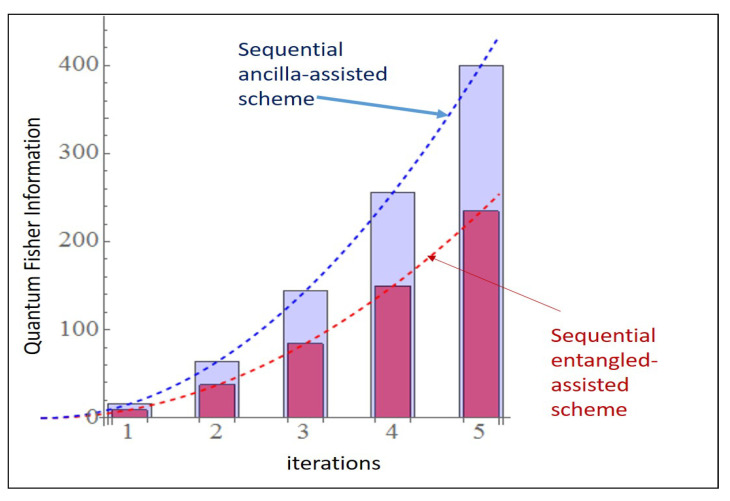
Comparison of two schemes in Figure 3 with multiple iterations for 4×4 the PPT state using Equations (Equation 22) and (Equation 23) combined as they are considered to be equivalent.

**Table 1 entropy-23-00685-t001:** Comparison of the two strategies in Figure 3 for all PPT states from reference [13]. The third column has been taken directly from [13] and the fourth column represents the QFI obtained using the sequential ancilla assisted strategy.

Without Decoherence			
**Dimension (d) of the System**	$F_{C}\left[\rho ,\hat{A}\right]$ **the Corresponding Classical Fisher Information**	$F_{Q}\left[\rho ,\hat{A}\right]$ **for Figure 4 (A) (** ***m*** ** = 1)**	$F_{Q}\left[\rho ,\hat{A}\right]$ **for Figure 5 (B) (** ***m*** ** = 2)**
3	8	8.0085	8.27623
4	8	9.3726	16.
5	8	9.3764	12.0935
6	8	10.1436	10.1436
7	8	10.1455	13.6191
8	8	10.6667	10.6667
9	8	10.6675	13.2849
10	8	11.0557	11.0557
11	8	11.0563	13.9923
12	8	11.3616	11.3616

## Data Availability

Not applicable.

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
