# Peer review of "Strategies for Positive Partial Transpose (PPT) States in Quantum Metrologies with Noise"

_entropy, 2021, doi:10.3390/e23060685_

Round 1

Reviewer 1 Report

The authors study quantum metrology with a family of bound entangled states presented in Ref. [12]. These are bipartite PPT entangled states that are more useful for metrology than separable states.

Authors try several metrological strategies that they call entangled assisted and sequential ancilla assisted strategy. In the first case, the phase shifts are applied to both parties. In the second case, one, two ore more phase shifts are applied to one of the two parties only. Authors also study the effect of noise on these scheme.

Fig. 7 shows the comparison between these two strategies. The sequential strategy performs better for several of the states.

I find the paper well written, very readable, about a very timely topic, thus. suggests its publication.

Comments:

- I think, for Fig 3 right, the Hamiltonian is

H_right=2 D_1 otimes Identity_2 + Identity_1 otimes Identity_2

since it does not act on system 2. For the left system, it is

H_left= D_1 otimes Identity_2 + Identity_1 otimes D_2

As far as I understand, the authors say that for H_right, the QFI is larger, while the two schemes need the same effort.

I think, the maximum for separable states, based on formula (7), is also larger. 

- Authors write that 

"Moreover, PPT states can enhance the ability for teleportation of other states [11] and they are found to be useful for quantum metrology [12]."

I suggest to add the following publication, since this would give a full review of this subfield in the sentence:

Activating hidden metrological usefulness, G Tóth, T Vértesi, P Horodecki, R Horodecki, Physical Review Letters 125, 020402

The paper above shows that there are PPT states that are not useful for metrology. However, their metrological capabilities can be activated, since two copies of these states become useful for metrology.

- Authors write

"QuantumFisher information possesses some interestingmathematical properties[16]. Indeed, it has been shown that, amongst all generalized analogues of quantum Fisher information, the quantum Fisher information is the largest for rank-2 density matrices and for operators that have zero diagonal elements in the eigenbasis of the density matrix [17]."

Indeed, in [17] it has been proven for certain special cases that the quantum Fisher information is the largest of all generalized analogues of quantum Fisher information, and verified numerically. In the following paper, it has been proven analytically:

Sixia Yu, Quantum Fisher Information as the Convex Roof of Variance

https://arxiv.org/abs/1302.5311

Eqs. (8,9,10) have been proven in two papers in appearing in parallel

https://arxiv.org/abs/1006.4366

Philipp Hyllus, WiesÅ‚aw Laskowski, Roland Krischek, Christian Schwemmer, Witlef Wieczorek, Harald Weinfurter, Luca Pezzé, Augusto Smerzi, Phys. Rev. A 85, 022321 (2012).

Géza Tóth, Multipartite entanglement and high-precision metrology, Phys. Rev. A 85, 022322 – Published 16 February 2012

Author Response

Dear sir, I am Arunava one of the authors of the manuscript. Thank you very much for your suggestions and for accepting our paper. We have made the following changes as you suggested : 

1. i) H_left= D_1 otimes Identity_2 + Identity_1 otimes D_2                     ------(A)

   H_right= D_1 otimes Identity_2 + D_1 otimes Identity_2                    -----(B)      since (B) does not act on system 2. In H_right the same term  D_1 otimes Identity_2 should be repeated.

       H_right =2 D_1 otimes Identity_2 + Identity_1 otimes Identity_2        ------(C)

Now (B) and (C) are the same as adding identity will not affect the relative quantum fisher information of separable states but most precisely eq. (C) will be

       H_right = D_1 otimes Identity_2 + D_1 otimes Identity_2 

As we described in the manuscript the local Hamiltonian terms must remain as it is as it will determine the fisher information for the separable states.

ii)  The fisher information for separable states in the case of H_right is the same as H_left as in both the cases the local operators have maximum eigenvalue = +1 and minimum eigenvalue  = -1 and thus using eq. 7 the fisher information for separable states = (+1-(-1))^2 + (+1-(-1))^2 = 4 + 4=8 so in both the cases the relative fisher information for separable states remains the same which is 8 but the QFI for PPT states become higher for H_right instead H_left and that is the advantage of using sequential strategy.

2. We have added more references including the ones that you suggested. 

Thank you for your kind consideration.

Reviewer 2 Report

Dear Editor

This paper presents an interesting analysis comparing ancilla- and sequential- assisted schemes for phase detection with positive partial transpose entangled states.

After a generic, but useful, introduction to quantum Fisher information, the two strategies are presented.

Then, various numerical examples for different cases are discussed, demonstrating the advantage of using the sequential scheme for increasing the Quantum Fisher Information.

The work is a little technical, but of interest for people working in the field of quantum metrology.

The presentation is sufficiently clear (I only suggest some linguistic check) and bibliography rather complete (I only suggest to expand a little it, in particular by quoting a few review papers more in the introduction)., figures are informative and readable (may be insets in Fig. 6 should be a little larger).

I suggest publishing it without further revisions, once the authors have  considered my  indications.

Author Response

Dear sir, I am Arunava one of the authors of the manuscript. Thank you very much for your suggestions and for accepting our paper. We have made the following changes as you suggested :

  1. we have made the insets in Fig. 6 a little larger.
  2. We extended the list of review papers and made some minor changes in the introduction part.
  3. The language has been improved upon.

Thank you for your kind consideration,